# ULTRAMEMV2: MEMORY NETWORKS SCALING TO 120B PARAMETERS WITH SUPERIOR LONG-CONTEXT LEARNING

**Zihao Huang**,[*] **Yu Bao**,[*] **Qiyang Min**,[*] **Siyan Chen, Ran Guo, Hongzhi Huang, Defa Zhu, Yutao Zeng**, **Banggu Wu**, **Xun Zhou**, **Siyuan Qiao**
Bytedance Seed
{huangzihao.notabot,baoyu.3302,minqiyang}@bytedance.com

## ABSTRACT

While Mixture of Experts (MoE) models achieve remarkable efficiency by activating only subsets of parameters, they suffer from high memory access costs during inference. Memory-layer architectures offer an appealing alternative with very few memory access, but previous attempts like UltraMem have only matched the performance of 2-expert MoE models, falling significantly short of state-of-the-art 8-expert configurations. We present UltraMemV2, a redesigned memory-layer architecture that closes this performance gap. Our approach introduces five key improvements: integrating memory layers into every transformer block, simplifying value expansion with single linear projections, adopting FFN-based value processing from PEER, implementing principled parameter initialization, and rebalancing memory-to-FFN computation ratios. Through extensive evaluation, we demonstrate that UltraMemV2 achieves performance parity with 8-expert MoE models under same computation and parameters but significantly low memory access. Notably, UltraMemV2 shows superior performance on memory-intensive tasks, with improvements of +1.6 points on long-context memorization, +6.2 points on multi-round memorization, and +7.9 points on in-context learning. We validate our approach at scale with models up to 2.5B activated parameters from 120B total parameters, and establish that activation density has greater impact on performance than total sparse parameter count. Our work brings memory-layer architectures to performance parity with state-of-the-art MoE models, presenting a compelling alternative for efficient sparse computation.

## 1 INTRODUCTION

Large language models (LLMs) have achieved remarkable success across NLP tasks, but their exponential growth in parameters and computational complexity presents significant challenges for resource-constrained deployment. Mixture of Experts (MoE)(Fedus et al., 2022; He et al., 2021; Liu et al., 2024; Muennighoff et al., 2024; Yuan et al., 2025) have emerged as a promising solution by selectively activating expert subsets, effectively decoupling parameter count from computational cost. Recent works (Muennighoff et al., 2024; Krajewski et al., 2024) show that MoE with 8 activated experts achieves optimal performance-efficiency trade-offs, significantly outperforming configurations with fewer experts. However, MoE inference suffers from high memory access costs due to expert routing overhead, particularly problematic when only a small fraction of tokens activate all experts.

Memory-layer architectures(Lample et al., 2019; Huang et al., 2024; Berges et al., 2024) offer an alternative sparse model with significantly less memory access. Unlike MoE's FFN-type expert, memory layers activate embeddings from large parameter table, enabling extremely slowly linear scaling of memory access with sequence length. The Over-tokenized Transformer (Huang et al., 2025) can also be viewed as a memory-layer architecture, where an n-gram router activates embeddings from a memory table that are subsequently added to the word embeddings. While architectures like Ultra-

---

[*]Equal contribution, Corresponding authors

Mem(Huang et al., 2024), current memory-layer State Of The Art(SOTA), demonstrate promising inference characteristics, they have only matched the performance of MoE with 2 activated experts, falling short of SOTA 8-expert configurations by a substantial margin.

This performance gap motivates our work. We introduce UltraMemV2, a redesigned memory-layer architecture that bridges the performance divide between embedding-based and expert-based sparse models. Our approach incorporates five key innovations: (1) **architectural integration**: tighter coupling between memory layers and Transformer blocks with memory layers in every block; (2) **simplified value expansion**: streamlined Implicit Value Expansion (IVE) using single linear projections; (3) **expert-like value processing**: adoption of PEER's FFN-based value computation (He, 2024); (4) **optimized initialization**: principled parameter initialization preventing training divergence; and (5) **computational rebalancing**: adjusted memory-to-FFN computation ratios.

Through comprehensive evaluation, we demonstrate that UltraMemV2 achieves performance parity with 8-expert MoE models while maintaining memory layer advantages. Notably, UltraMemV2 shows superior performance on memory-intensive tasks including long-context memorization (+1.6 points), multi-round memorization (+6.2 points), and in-context learning (+7.9 points). We validate scalability up to 2.5B activated parameters with 120B total parameters, and establish that activation density (top-m values) has greater impact on performance than total sparse parameter count.

In summary, our work makes three primary contributions: (1) Architectural advancement: We present the first memory-layer architecture competitive with state-of-the-art 8-expert MoE models, closing a significant performance gap in sparse model research. (2) Comprehensive analysis: We provide detailed ablation studies and comparative analysis revealing when and why UltraMemV2 outperforms MoE, particularly on memory-intensive tasks, while identifying trade-offs in different training phases. (3) Scalability validation: We demonstrate UltraMemV2's effectiveness at scale and establish design principles for activation density versus parameter count trade-offs, providing guidance for future memory-layer architectures.

## 2 RELATED WORK

**MoE Architecture** The concept of MoE was first introduced by Shazeer et al. (2017). Since then, numerous studies (Fedus et al., 2022; Dai et al., 2024; Rajbhandari et al., 2022; Jiang et al., 2024) have been conducted to improve its performance and efficiency. During this period, the general perception is that appropriately using smaller experts but activating a greater number can enhance the performance of MoE, typically activating two experts. Krajewski et al. (2024) systematically studied the influence of expert size and the number of activations, which is called "granularity". They found that when the granularity was 8, MoE achieved the best performance and was significantly better than 2. The same conclusion was also discovered by OLMoE(Muennighoff et al., 2024). Resent MOEs in the industrial sector (DeepSeek-V3(Liu et al., 2024), Qwen3(Yang et al., 2025), dots.llm1(Huo et al., 2025)) have all adopted this structure. However, they still face challenges in inference, such as high memory access costs and long inference latency, especially when dealing with large-scale models.

**Memory Layer Architecture** The idea of a memory layer was first explored by Lample et al. (2019) with the introduction of the Product Key Memory (PKM). By activating embeddings instead of expert, PKM aimed to expand the model's parameters while maintaining similar computation and less memory access. Subsequently, several improvements have been made to PKM. For example, Kim & Jung (2020) introduced a concept similar to shared experts in MoE, allowing PKM and MLP to operate in parallel. Csordás et al. (2023) made a slight modification to PKM by removing the Softmax operation. He (2024) proposed PEER, which improved the activation of values in PKM by using an FFN with one inner dimension. Memory+(Berges et al., 2024) also made some improvements to the memory layer architecture. However, most of these memory layer architectures have only managed to match the performance of MoE models with **one** activated experts. The UltraMem(Huang et al., 2024) was an attempt to address the limitations of existing memory layer architectures. It incorporated Tucker Decomposed Query Key retrieval (TDQKR) and Implicit Value Expansion (IVE) to improve model performance while maintaining inference latency. However, it still can only matchs the performance of MoE with **two** activated experts. UltraMemV2 builds on the previous work and aims to overcome these limitations. By introducing several innovative improvements, UltraMemV2

can achieve comparable performance to MoE models with **eight** activated experts, filling the gap in the current research on memory layer architectures.

# 3 APPROACH

## 3.1 PRELIMILARY

Memory layers are structures designed to expand model capacity without a proportional increase in computational cost. We briefly review three common architectures: MoE, PKM and UltraMem.

**MoE Layer**, as shown in Figure 6(a), utilizes a gating mechanism to selectively activate a subset of parameters. Given a hidden state $\mathbf{x} \in \mathbb{R}^{D_{\text{in}}}$, a gate with parameters $\mathbf{K} \in \mathbb{R}^{N \times D_{\text{in}}}$ computes routing scores for $N$ experts.

$$\mathbf{s} = \mathbf{x} \times \mathbf{K}^T \tag{1}$$

The `Top-M` function selects the indices $\mathcal{I}$ of the $m$ experts with the highest scores. These indices are used to retrieve the corresponding parameters from the expert pool, which consist of Pre-values $\mathbf{U} \in \mathbb{R}^{N \times D_{in} \times D_{inner}}$ and Values $\mathbf{V} \in \mathbb{R}^{N \times D_{out} \times D_{inner}}$. The final output $\mathbf{o}$ is the combination of the activated parameters, often weighted by the gating scores $s_i$ for $i \in \mathcal{I}$. A common formulation is:

$$\mathbf{o} = \sum_{i \in \mathcal{I}} s_i \cdot (\text{SiLU}(\mathbf{x}\mathbf{U}_i) \times \mathbf{V}_i^T) \tag{2}$$

where $\mathbf{U}_i$ and $\mathbf{V}_i$ are the parameters for the $i$-th expert.

**PKM Layer**, illustrated in Figure 6(b), employs key factorization to create a large memory from smaller, more efficient key-value sets. The input hidden state $\mathbf{x}$ is first projected to a query $\mathbf{q}$ via linear layer $q_{row}$ and $q_{col}$. This query is then used to compute scores against multiple, smaller sets of factorized keys, row keys $\mathbf{K}_{row} \in \mathbb{R}^{N \times D_k}$ and column keys $\mathbf{K}_{col} \in \mathbb{R}^{N \times D_k}$ to retrieve values from $N^2$ value pool by Product Quantizaion (PQ)(Jegou et al., 2010). The scores from these factorized components are aggregated to identify the most relevant memory value in the full memory space. A `Top-M` function selects the indices $\mathcal{I}$ and scores $\mathbf{S}_{grid}$ for the best-matching memory entries.

$$\mathbf{s}_{row} = \sigma_{\text{TopM}}(\mathbf{K}_{row}q_{row}(\mathbf{x})), \quad \mathbf{s}_{col} = \sigma_{\text{TopM}}(\mathbf{K}_{col}q_{col}(\mathbf{x})), \tag{3}$$

$$\mathbf{S}_{grid} = \sigma_{\text{TopM}}(\mathbf{s}_{row} + \mathbf{s}_{col}^\top). \tag{4}$$

The final output is a weighted sum of the corresponding values $\mathbf{V}_i$ retrieved from the value memory.

$$\mathbf{o} = \sum_{i \in \mathcal{I}} \mathbf{S}_{grid}^i \mathbf{V}_i \tag{5}$$

Commonly PKM use multi-head trick, which is similar to Multi-head attention(Vaswani et al., 2017), we omit this operation in the above formula description for brevity. This factorization allows for a much larger memory capacity than could be addressed by a single, monolithic key matrix, while keeping the number of parameters manageable.

**UltraMem layer** is illustrated in Figure 6(c). It typically consists of keys $\mathbf{K}_{row}, \mathbf{K}_{col} \in \mathbb{R}^{n, D_k, r}$, tucker core $\mathbf{C} \in \mathbb{R}^{r, r}$ and values $\mathbf{V} \in \mathbb{R}^{n^2, D_v}$. Given a hidden state $\mathbf{x} \in \mathbb{R}^{D_i}$, scores for activated values are computed by Tucker Decomposed Query-Key Retrieval (TDQKR):

$$\mathbf{S}_{row} = \mathbf{K}_{row}q_{row}(\mathbf{x}), \qquad\qquad \mathbf{S}_{col} = \mathbf{K}_{col}q_{col}(\mathbf{x}), \tag{6}$$

$$\mathbf{S}_{grid} = \sigma_{\text{TopM}}(\mathbf{S}_{row}^\top \times \mathbf{C} \times \mathbf{S}_{col}), \tag{7}$$

where $q_{row}, q_{col} : \mathbb{R}^{D_i} \to \mathbb{R}^{D_k}$ linearly convert the dimension of the input to $D_k$, $\sigma_{\text{TopM}}(\cdot)$ selects the top-$m$ scores and set the remaining scores to negative infinity. Then weighted sum pooling with Implicit Value Expansion (IVE) is conducted to generate the final output of memory layer:

$$\hat{\mathbf{s}} = \text{Shuffle}(\text{vec}(\mathbf{S}_{grid})), \tag{8}$$

$$\mathbf{o} = \tilde{\mathbf{V}}^\top \times \hat{\mathbf{s}} = \sum_p \tilde{\mathbf{V}}_p^\top \times \hat{\mathbf{s}}_p = \sum_p \mathbf{W}_p^\top \left(\mathbf{V}^\top \times \hat{\mathbf{s}}_p\right), \tag{9}$$

where operation `Shuffle` aims to eliminate some unnecessary index topology prior introduced by row and column scoring, $\hat{\mathbf{s}}_p$ represents the scores corresponding to $p$-th virtual memory block, and $\mathbf{W}_p \in \mathbb{R}^{D_v, D_i}$ is linear projector for $p$-th virtual memory block. We have overlooked cumbersome operations and only present the key ones here.

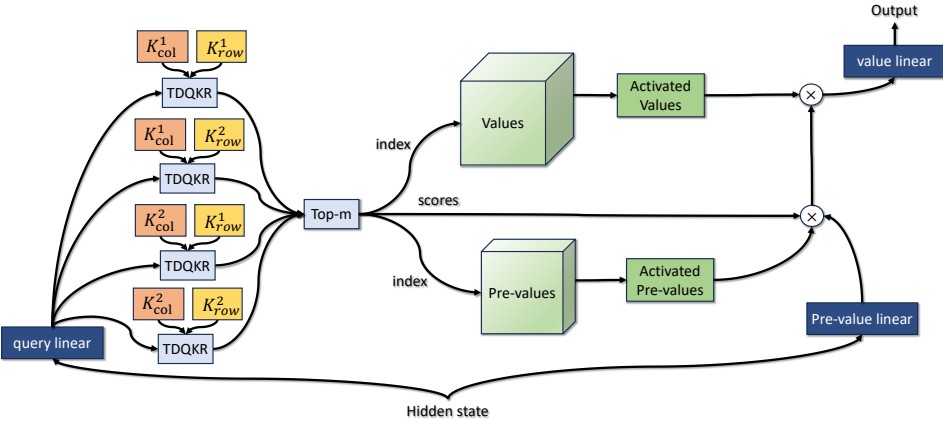

Figure 1: Overall structure of UltraMemV2 layer.

## 3.2 OVERALL STRUCTURE

We present an improved structure called UltraMemV2, shown in Figure 1. Compared to Ultra-Mem (Huang et al., 2024), we highlight the improvements in the following:

1) Every transformer block contains an FFN layer and an UltraMemV2 layer.

2) The multiple linear layers in Implicit Value Expansion (IVE) are removed and use only a single linear layer. Meanwhile, we use separate queries for each tucker rank.

3) PEER (He, 2024) is adopted, by which the embedding value is changed to an FFN with one inner dimension.

4) We improve the initialization of the parameters in the new structure.

5) We adjust the proportion of the calculation for memory layer.

## 3.3 DIFFERENT VIEW IN IMPLICIT VALUE EXPANSION

Given a set of row and column keys $\mathbf{K}^i_{row}, \mathbf{K}^j_{col}, i, j \in [1, 2, ..., h]$, the $\mathbf{S}_{grid}$ in Equation 8 is obtained by

$$\mathbf{S}^i_{row} = \mathbf{K}^i_{row} q_{row}(\mathbf{x}), \mathbf{S}^i_{col} = \mathbf{K}^j_{col} q_{col}(\mathbf{x}), \tag{10}$$

$$\mathbf{S}_{grid} = \sigma_{\text{TopM}}([\mathbf{S}^{1\top}_{row} \mathbf{C}^{1,1} \mathbf{S}^1_{col}, \mathbf{S}^{1\top}_{row} \mathbf{C}^{1,2} \mathbf{S}^2_{col}, ..., \mathbf{S}^{i\top}_{row} \mathbf{C}^{i,j} \mathbf{S}^j_{col}, ..., \mathbf{S}^{h\top}_{row} \mathbf{C}^{h,h} \mathbf{S}^h_{col}]). \tag{11}$$

For a standard multi-head structure that requires $h^2$ heads, generally $h^2$ row and column keys are needed. IVE introduces shared key pairs, so only $h$ row and column keys are needed to achieve $h^2$ heads. Therefore, IVE has actually implemented a multi-head mechanism, there is no need to explicitly define a multi-head mechanism like that in Product Key Memory (PKM) (Lample et al., 2019).

Considering that IVE adds a linear layer for each head to remap, but mapping different embeddings to corresponding linear layers will increase additional non-computational operations, affecting the inference speed. Meanwhile, we find that if the linear layer is shared and the saved parameters are added to the FFN, better results can be achieved. This indicates that the parameter efficiency of nonshared linear layers is actually not high. As a result, Equation 9 is modified as:

$$\mathbf{o} = \tilde{\mathbf{V}}^\top \times \hat{\mathbf{s}} = \tilde{\mathbf{V}}^\top \times \hat{\mathbf{s}} = \mathbf{W}^\top \left( \mathbf{V}^\top \times \hat{\mathbf{s}} \right). \tag{12}$$

## 3.4 MILLION OF 1-INNER-DIM EXPERTS INSTEAD OF EMBEDDINGS

PEER(He, 2024) uses FFN with one inner dimension replacing the value. Given pre-value weight matrix $\mathbf{P} \in \mathbb{R}^{n^2, D_p}$, the final output is

$$\mathbf{o} = \mathbf{W}^\top \left( \mathbf{V}^\top \times (\sigma(\mathbf{P}\mathbf{x}) \otimes \hat{\mathbf{s}}) \right), \tag{13}$$

where $\mathbf{x}$ is the input, $\sigma$ is the activate function. Consider a standard SwiGLU FFN, given $W_1, W_2 \in \mathbb{R}^{H,N}, W_3 \in \mathbb{R}^{N,H}$,

$$\mathbf{o} = \mathbf{W_3}^\top \times (\mathbf{W_2 x} \otimes \sigma(\mathbf{W_1 x})). \tag{14}$$

We find that PEER is very similar to SwiGLU FFN(Shazeer, 2020), where $\mathbf{V}$, $\sigma(\mathbf{Px})$ and $\hat{\mathbf{s}}$ corresponds to $\mathbf{W_3}$, $\mathbf{W_2 x}$ and $\sigma(\mathbf{W_1 x})$, respectively. It should be noted that $\hat{\mathbf{s}}$ comes from Equations 11 and 8, where `TopM` can be seen as an activate function. We argue that empirically, applying the activation function to two parallel results simultaneously is lossy. Therefore, we decide to remove the activation function in FFN based on PEER, leads to the final output as

$$\mathbf{o} = \mathbf{W}^\top \left( \mathbf{V}^\top \times ((\mathbf{Px}) \otimes \hat{\mathbf{s}}) \right). \tag{15}$$

For the sake of simplicity, this change will be uniformly abbreviated as PEER in this paper.

## 3.5 Improved initialization

When using a normal distribution to initialize the UltraMemV2 layer, the standard deviation must be carefully designed; otherwise, the training process is very prone to divergence. The selection criteria for the initialization standard deviation are: (1) After initialization, the variance of the output activation of the memory layer does not diverge with the increase in the number of layers; (2) The variance of the output activation should not be too large. Considering that memory can be regarded as an enhancement of FFN to a certain extent, we make the initialization activation variance of the memory layer consistent with that of FFN. The specific derivation process of the initialization variance can be found in the Appendix B.

## 4 Experiments

This section provides a comprehensive experimental validation of the UltraMemV2 architecture. Our evaluation encompasses three primary objectives:

1. Performance Parity Validation: We demonstrate that UltraMemV2 achieves comparable performance to state-of-the-art 8-expert MoE models, thereby bridging the substantial performance gap that has historically limited memory-layer architectures.

2. Architectural Advantage Analysis: We validate UltraMemV2's superior performance on memory-intensive tasks, with particular emphasis on long-context memorization, multi-round memorization, and in-context learning capabilities. However, we also identify certain limitations, including reduced effectiveness during early training phases and potential performance trade-offs in specific reasoning tasks compared to MoE.

3. Component Effectiveness Assessment: Verifying the effectiveness of core improvements through ablation studies, while reducing hyperparameter configuration requirements and simplifying the training pipeline.

**Training Data** Our experiments utilize both proprietary and open-source datasets. The proprietary training corpus comprises 3.9T tokens for PreTraining (PT) and 500B high-quality long context tokens for continued training (CT). For open-source comparisons, we employ the 1T token dataset from OLMoE to ensure fair evaluation against existing baselines.

**Evaluation Benchmarks** We conduct evaluation across diverse benchmark suites encompassing both proprietary and open-source assessments. These include comprehensive evaluations of math, code, reasoning and knowledge capabilities. Additionally, we evaluate long-context performance through specialized benchmarks measuring long-context memorization, long-context reasoning, needle in a haystack, and long-document retrieval capabilities. Detailed specifications of all evaluation datasets are provided in the Appendix C.

## 4.1 Compare to MoE

We evaluate UltraMemV2 against both proprietary and open-source baselines across multiple training stages and benchmarks. For proprietary models, we compare against SeedMoE variants with

different parameter configurations. For open-source models, we benchmark against OLMoE, Memory+, and UltraMem architectures.

**Training Protocol**: Proprietary models undergo a two-stage training process: (1) pretraining (PT) on 1.6T tokens, followed by (2) continued training (CT) on 250B high-quality tokens. Selected models are further pretrained to 3.9T tokens with additional 32K context CT using 500B tokens. Open-source models undergo 500B or 1T tokens.

**Model Configurations**: For the proprietary model, UltraMemV2-2.5B/120B-top256 activates 2.5B parameters from 120B sparse parameters with 256 activated values per UltraMemV2 layer. UltraMemV2-2.5B/60B-top768 uses 768 activated values from 60B sparse parameters. We constrain row/column `TopM` to 128 to avoid quadratic intermediate variable explosion. For the open-source model, we make sure the same computation and parameters. The OLMoE has 64 experts, and each token activates 8 experts. Memory+ and UltraMem contains 4 memory layers and each memory layer has 2 heads with $\text{TopM} = 80$. UltraMemV2 contains 20 memory layers, which has 1 head with $\text{TopM} = 32$. All three memory-layer-based model activate same amount of value embeddings. Detail model hyperparameters is shown in Appendix D.

**Proprietary Model Comparison** Table 1 presents comprehensive evaluation results across Open-Bench and HardBench benchmarks at different training stages. Table 2 demonstrates UltraMemV2's capabilities on 32K long-context tasks. We observe several key findings:

1. Training Stage Dependency: UltraMemV2 exhibits distinct performance characteristics across PT and CT phases. After 1.6T PT, UltraMemV2-2.5B/60B-top768 underperforms SeedMoE-2.5B/60B on mathematical reasoning, coding, and reasoning tasks. However, following 250B CT, UltraMemV2 achieves competitive or superior performance across all metrics, suggesting enhanced sensitivity to high-quality data and learning rate decay schedules.

2. Scaling behavior: After extended training (3.9T PT + 500B CT), UltraMemV2-2.5B/60B-top768 shows marginal improvements over SeedMoE-2.5B/30B on OpenBench but demonstrates clear advantages on HardBench. The diminishing returns may reflect general scaling limitations rather than architecture-specific issues, though this warrants further investigation with larger parameter budgets.

3. Architecture trade-offs: Comparing UltraMemV2-2.5B/60B-top768 versus UltraMemV2-2.5B/120B-top256, we find that increasing activated values per layer (top768 vs top256) yields better performance than increasing total sparse parameters (60B vs 120B). This suggests activation density is more important than sparse parameter count, though higher activation increases inference latency.

4. Long-Context Performance: UltraMemV2 shows substantial improvements in memory-intensive tasks (long-context memorization: 23.5 vs 21.9, multi-round memorizing: 31.2 vs 25.0, in-context learning: 29.5 vs 21.6) compared to SeedMoE. Performance variations in Key-value retrieval and Multi-hop reasoning appear attributable to architectural differences rather than parameter count disparities.

Table 1: Performance comparison of different models across various benchmarks. Models are grouped by training tokens with consistent background colors.

| Model | Training tokens | Training loss | Eval loss | Openbench | | | | | Hardbench | | | | |
|---|---|---|---|---|---|---|---|---|---|---|---|---|---|
| | | | | knowledge | reasoning | Math | code | All | knowledge | reasoning | Math | code | All |
| SeedMoE-2.5B/30B | 1.6T PT | 1.812 | 1.900 | 70.6 | 71.8 | 62.8 | 43.1 | 60.5 | 24.8 | 49.8 | 15.4 | 10.1 | 23.3 |
| | +250B CT | 1.165 | 1.846 | 77.0 | 77.9 | 71.3 | 50.7 | 67.6 | 34.2 | 53.9 | 18.9 | 15.2 | 27.4 |
| | 3.9T PT | 1.794 | 1.894 | 73.3 | 74.0 | 65.6 | 45.6 | 63.1 | 25.9 | 51.8 | 16.5 | 10.9 | 23.5 |
| | +500B CT | 1.049 | 1.837 | 79.5 | 79.3 | 75.1 | 56.6 | 70.7 | 39.2 | 53.8 | 23.1 | 19.3 | 30.3 |
| SeedMoE-2.5B/60B | 1.6T PT | 1.793 | 1.869 | 73.1 | 72.5 | 64.4 | 43.4 | 61.6 | 27.4 | 52.4 | 15.5 | 10.0 | 23.1 |
| | +250B CT | 1.123 | 1.796 | 79.1 | 76.9 | 71.2 | 54.4 | 68.1 | 35.6 | 56.7 | 21.5 | 17.3 | 29.2 |
| UltraMemV2-2.5B/120B-top256 | 1.6T PT | 1.752 | 1.835 | 73.6 | 70.8 | 61.3 | 39.5 | 60.4 | 26.7 | 44.7 | 14.5 | 9.2 | 21.2 |
| | +250B CT | 1.066 | 1.803 | 80.7 | 76.4 | 71.8 | 52.4 | 68.3 | 35.5 | 54.4 | 20.4 | 16.7 | 27.9 |
| UltraMemV2-2.5B/60B-top768 | 1.6T PT | 1.769 | 1.855 | 75.1 | 71.5 | 61.2 | 40.4 | 60.3 | 26.9 | 51.7 | 14.4 | 9.5 | 22.2 |
| | +250B CT | 1.100 | 1.770 | 80.3 | 76.8 | 72.5 | 55.8 | 69.1 | 35.6 | 56.2 | 22.7 | 16.2 | 30.0 |
| | 3.9T PT | 1.748 | 1.847 | 76.4 | 74.2 | 62.6 | 45.6 | 62.8 | 27.8 | 53.4 | 16.5 | 11.8 | 24.5 |
| | +500B CT | 0.975 | 1.784 | 81.7 | 79.0 | 73.9 | 56.9 | 70.7 | 38.9 | 57.5 | 23.8 | 19.4 | 31.7 |

Table 2: Performance comparison on long-context tasks.

| Model | Long-context memorizing | Multi-round memorizing | In-context learning | Reasoning | Find Needle | Key-val retrieval | Multi-hop reasoning | All |
|---|---|---|---|---|---|---|---|---|
| SeedMoE-2.5B/30B | 21.9 | 25.0 | 21.6 | 6.7 | 96.5 | 41.3 | 34.8 | 35.4 |
| UltraMemV2-2.5B/60B-top768 | 23.5 | 31.2 | 29.5 | 7.7 | 97.0 | 57.1 | 17.7 | 37.7 |

**Open-Source Model Comparison** Table 3 evaluates UltraMemV2 against open-source alternatives under controlled parameter budgets. UltraMemV2 significantly outperforms memory-based architectures (Memory+, UltraMem) while achieving competitive performance with OLMoE across 227M/1.2B and 1B/7B configurations. This validates UltraMemV2's effectiveness relative to current state-of-the-art expert-routing MoE approaches.

Table 3: Comprehensive performance comparison across different open sourced models and benchmarks.

| Model | Training | | Eval | Evaluation Benchmarks | | | | | | | | | |
|---|---|---|---|---|---|---|---|---|---|---|---|---|---|
| | Tokens | Loss | Loss | ARC-C | ARC-E | Common-senseQA | Hellas-wag | MMLU-var | Open-bookQA | PIQA | SCIQ | Wino-grande | All |
| OLMoE-227M/1.2B | 500B | 2.482 | 2.845 | 34.1 | 65.4 | 42.2 | 58.9 | 33.0 | 35.6 | 73.8 | 89.4 | 58.9 | 54.6 |
| Memory+-227M/1.2B | 500B | 2.566 | 2.920 | 32.4 | 66.5 | 39.2 | 53.6 | 32.9 | 35.0 | 72.2 | 87.3 | 56.3 | 52.8 |
| UltraMem-227M/1.2B | 500B | 2.528 | 2.885 | 35.8 | 66.7 | 42.3 | 55.2 | 32.8 | 36.2 | 73.4 | 87.4 | 54.7 | 53.8 |
| UltraMemV2-227M/1.2B | 500B | 2.500 | 2.853 | 31.8 | 66.7 | 43.0 | 58.1 | 34.0 | 37.4 | 73.1 | 89.3 | 56.4 | 54.4 |
| OLMoE-1B/7B | 1T | 2.262 | 2.631 | 43.1 | 74.6 | 49.3 | 71.5 | 39.0 | 43.0 | 78.6 | 92.7 | 66.2 | 62.1 |
| UltraMemV2-1B/7B | 1T | 2.266 | 2.628 | 44.5 | 74.7 | 50.0 | 71.4 | 39.7 | 41.0 | 77.8 | 93.1 | 64.1 | 61.8 |

These results demonstrate UltraMemV2's viability as a competitive alternative to current MoE paradigms.

## 4.2 STRUCTURE ABLATION

### 4.2.1 PEER

We investigate the impact of using PEER (He, 2024) who first suggested replacing value embedding with an FFN having a 1-dimensional inner layer. Our ablation study on UltraMemV2-430M/5B compares two settings. The first, "Baseline", is configured with $N = 789$ keys, a value dimension $D_v = 288$, and TopM = 48. The second, "PEER", differs by using $N = 558$ keys, a pre-value dimension $D_p = 288$ (while $D_v$ remains 288), and TopM = 24. Under this configuration, the calculation and memory access of the memory layer are completely consistent. As illustrated in Figure 2, which presents the training loss and downstream accuracy, PEER demonstrates a significant advantage over the standard value embedding approach, noticed that the parameters involved in processing the activated top-$m$ values is kept constant. This highlights the parameter efficiency and effectiveness of the FFN-based value processing proposed by PEER.

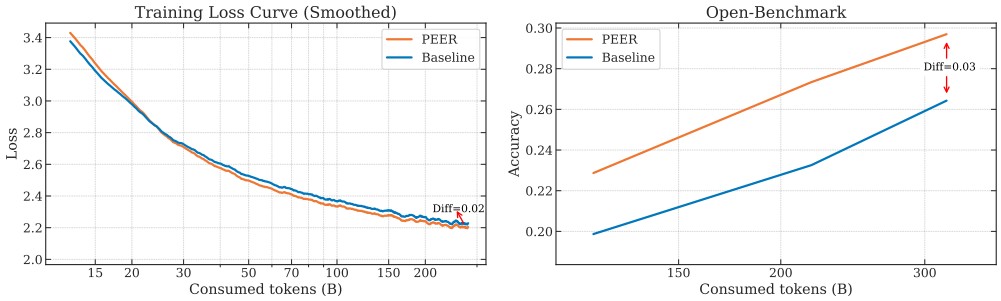

Figure 2: PEER ablation study on UltraMemV2-430M/5B. Training loss (left) and Open-Benchmark accuracy (right) comparing PEER with Baseline using value embedding.

### 4.2.2 HOW LARGE DO $D_v$ AND $D_p$ NEED TO BE?

First, we set pre-value dimention $D_p$ equal to value dimention $D_v$ to explore the overall dimension configuration. In Table 9, the memory computational proportion 19.5% has $D_p = D_v = H/8$.

Based on this configuration, we also tried $D_p = D_v = H/4$, while ensuring the same number of parameters and the same computational load.

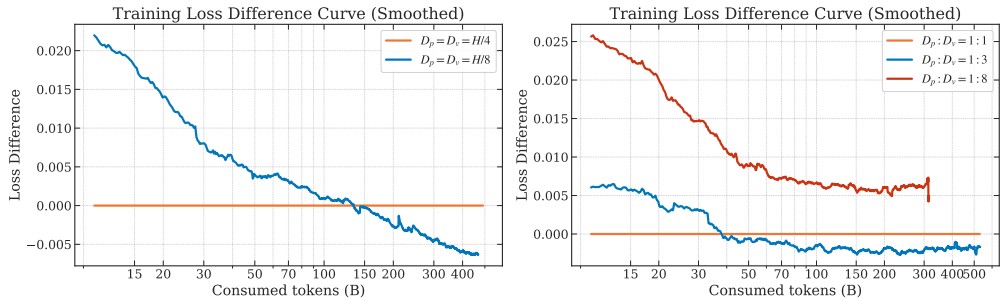

Figure 3: Left: ablation on the sum of $D_v$ and $D_p$. Right: ablation on the proportion of $D_v$ and $D_p$.

Figure 3(left) compares the loss performance. The results indicate that the smaller the sum of $D_v$ and $D_p$, the more refined the division of memory. To maintain the same sparsity, larger key number $N$ and `TopM` are required, which increases the number of combinations in the UltraMem part and thus leads to better performance. However, larger $N$ and `TopM` will slow down the training and inference process, therefore, we did not further reduce the $D_v$ and $D_p$.

Next, we explored how to set $D_v$ and $D_p$ respectively. Based on the configuration with MCP = 17.0% in Table 9, we kept the sum of $D_v$ and $D_p$ unchanged, and further increased $D_v$ and reduced $D_p$. We tried the following two configurations in Table 8. Figure 3(right) illustrates the loss differences. Based on the loss result, we set $D_p : D_v = 1 : 3$.

### 4.2.3 ULTRAMEMV2 LAYER NUMBER

Under normal circumstances, MoE is present in each layer. This will increase the participation of sparse parameters in the model pipeline. UltraMemV2 is distributed in the model at fixed-layer intervals. We conduct an experiment in which the number of UltraMemV2 layers $L_m$ is gradually increased. Meanwhile, we make sure that the `TopM` $\times L_m$ and the total computation is fixed. The UltraMemV2 in this section is based on Seed-MoE and replaces MoE with SwiGLU FFN, inserting UltraMemV2 layers at regular intervals (fixed intervals of 1 when each layer is available).

Figure 4 shows the validation loss and open benchmark changes for UltraMemV2-430M/5B when inserting layers 2, 5, 10, and 20. We observe that validation loss quickly become indistinguishable when increasing the number of insertions of the UltraMemV2 layer, but sustained gains are observed on the Open Benchmark (obtain +2.3, +0.9, +1.1). Memory+(Berges et al., 2024) also did a similar experiment, but with different conclusions. We speculate that the reason is that we keep FFN in each layer, while they directly replace FFN with the Memory+ layer.

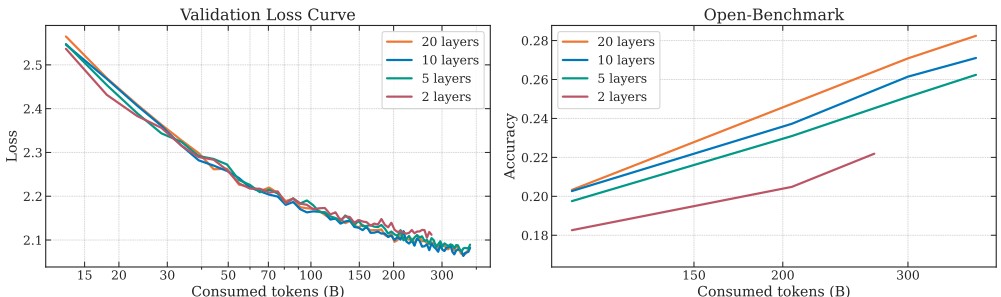

Figure 4: Effect of UltraMem layer number on training dynamics and performance. Validation loss (left) and open-benchmark accuracy (right) for UltraMemV2-430M/5B with varying numbers of UltraMemV2 layers (2, 5, 10, 20) under fixed computational budget. Although there is no obvious gain in the validation set loss after adding a certain number of layers, the model with more UltraMemV2 layers performs better in downstream tasks.

We then perform continued training (CT) on UltraMem and MoE, but we observed that in the Ul-traMemV2, the CT gains are smaller. Specifically, we pretrain the model with 1.6T tokens under constant learning rate, then continue training with 250B tokens under cosine decay learning rate. As shown in Table 4, the average score of UltraMem-430M/5B with 4 UltraMem layers improves by 5.7 points after CT, but MoE improves by 7.8 points. In general, the larger the model, the less points it improves after CT, so UltraMem-430M/5B should have at least a gain of more than 7.8 points to be normal, which affects the effect of UltraMem post training. We find that the key to solving this problem lies in the number of UltraMem layers. When each transformer block has UltraMem layer, the gains of UltraMemV2 and Seed-MoE become consistent on 1.25B/12.5B and 2.5B/25B models.

Table 4: Performance improvements after CT. "*" represents model structure based on older version of Seed-MoE. Evaluation sets not included in the category of reasoning, mathematics, code and knowledge are listed separately, including DROP(Dua et al., 2019), AGIEval(Zhong et al., 2023).

| Model | Reasoning | Math | Code | Knowledge | DROP | AGIEval | Average |
|---|---|---|---|---|---|---|---|
| Seed-MoE*-2.5B/25B | +4.6 | +10.2 | +11.5 | +5.2 | +7.6 | – | +7.8 |
| UltraMemV2*-4L-430M/5B | +3.9 | +13.1 | +4.1 | +5.1 | +2.6 | – | +5.7 |
| Seed-MoE-1.25B/12.5B | +4.9 | +11.3 | +11.5 | +6.7 | – | +8.4 | +7.5 |
| UltraMemV2-1.25B/12.5B | +4.8 | +15.6 | +10.2 | +5.7 | – | +9.9 | +8.3 |
| Seed-MoE-2.5B/25B | +6.1 | +8.6 | +9.0 | +5.8 | – | +8.2 | +7.1 |
| UltraMemV2-2.5B/25B | +5.7 | +12.3 | +11.2 | +5.7 | – | +8.5 | +8.0 |

### 4.2.4 OTHER ABLATIONS

Appendix E shows more ablations, which reveal several key insights for UltraMemV2: Architectural simplifications (single value projector, separate queries per Tucker rank, single head) consistently improve performance. An optimal 17% memory computational proportion is identified. Auxiliary losses, including Tucker core penalty and balance loss, are found to be ineffective. Layer-wise memory sharing (S9-Ring, S6-Block) significantly boosts performance, offering scalable options. Lastly, while decaying learning rates for value parameters provide early gains, a constant rate achieves better final performance over extended training.

### 4.3 TRAINING AND INFERENCE TIME

We trained UltraMemV2-2.5B/60B-top768 and SeedMoE-2.5B/60B on 1008 NVIDIA Hopper GPUs with a batch size of 16M tokens per step. Both models achieve comparable training through-put: UltraMemV2 reaches 265B tokens/day while Seed-MoE reaches 262B tokens/day. However, the two architectures exhibit different resource constraints. UltraMemV2 is memory-bound, con-suming more bandwidth due to atomic memory operations, while MoE is compute-bound. Despite these different bottlenecks, they achieve similar overall training efficiency on current GPUs. This suggests that hardware with improved memory efficiency, particularly optimizations for atomic op-erations, would disproportionately benefit UltraMemV2, enabling further training speedup.

We evaluated the inference time and memory access of UltraMemV2-10B/200B-top768 and Seed-MoE-10B/200B across different batch sizes in Figure 5. The results show that UltraMemV2 consis-tently achieves significantly lower memory access than MoE at commonly used batch sizes, while delivering up to $2\times$ faster inference speed compared to Seed-MoE. We observe that smaller batch sizes yield even greater speedups; however, for practical considerations, we limit our evaluation to batch sizes of 32 and above. These results demonstrate the inference advantages enabled by Ultra-MemV2's extreme sparsity.

### 4.4 DISCUSSION

**Integrating UltraMemV2 During Mid-Training** While our experiments focus on training Ultra-MemV2 from scratch, the architecture is amenable to integration at later training stages. Memory layers can be straightforwardly added during mid-training or post-training phases. A practical ap-proach is to insert UltraMemV2 layers into an existing model and use zero initialization for either

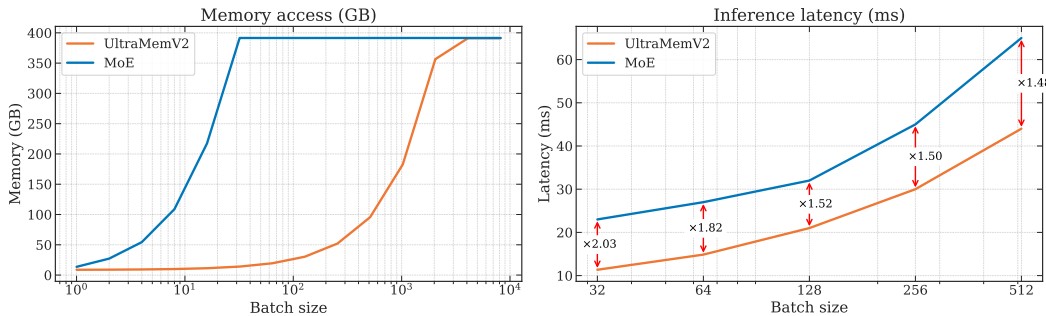

Figure 5: Changes in memory access and inference latency as the batch size varies. UltraMemV2 and MoE both has 10B activated parameters and 200B paremeters. Inference latency is evaluated on NVIDIA Hopper GPUs.

the value projection or a gating scalar. This ensures smooth integration without disrupting the pre-trained weights, allowing the memory layers to be gradually incorporated as training progresses. Recent work has demonstrated promising applications of this approach. For instance, Lin et al. (2025) shows that integrating memory layers during fine-tuning can mitigate catastrophic forgetting. We believe that incorporating UltraMemV2 during continued training (CT) or fine-tuning phases represents a valuable research direction, particularly for scenarios requiring efficient model adaptation or knowledge retention.

**Combining UltraMemV2 with MoE Architectures** UltraMemV2 and MoE are not mutually exclusive, they can potentially be combined within a single architecture. One straightforward approach is to use UltraMemV2 layers alongside traditional MoE layers within the same model, leveraging the complementary strengths of both architectures. However, converting a fully pretrained MoE model to UltraMemV2 is more challenging due to the fundamental differences in their architectural designs. The most practical path forward is to design hybrid architectures from the outset or to introduce UltraMemV2 components during mid-training, as discussed above. We leave detailed exploration of such hybrid architectures as promising future work.

## 5 CONCLUSION

In this paper, we developed UltraMemV2, a new type of architecture that, for the first time, performs as well as the top-tier 8-expert MoE models. By placing our new memory layers in every part of the model, we show that this approach is a solid new option for building large and efficient AI.

Our work has a few key takeaways. First, UltraMemV2 matches the performance of powerful MoE models on standard tests. Second, it's particularly good at tasks that require a great memory, like long-document understanding and multi-turn conversations, where it significantly outperforms MoE. Third, we found that we could simplify the training process, removing the need for extra complex settings. Finally, we learn an important design lesson: It is better to activate more values than to simply increase the number of sparse parameters.

However, there are some limitations to be aware of. UltraMemV2 gets off to a slow start; it doesn't perform as well as MoE models early in its training and needs a lot of high-quality data to catch up. Its best performance also depends on putting a memory layer in every single block of the model.

In summary, UltraMemV2 validates the potential of memory-layer architectures for building efficient and powerful large-scale models. Future work should focus on improving its early-stage training dynamics and further exploring architectural trade-offs for diverse downstream applications.

THE USE OF LARGE LANGUAGE MODEL

The Large Language Model is only used to polish the content of this paper, correct grammatical errors, and make the paper more coherent and easy to understand.

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

## A   PRELIMINARY

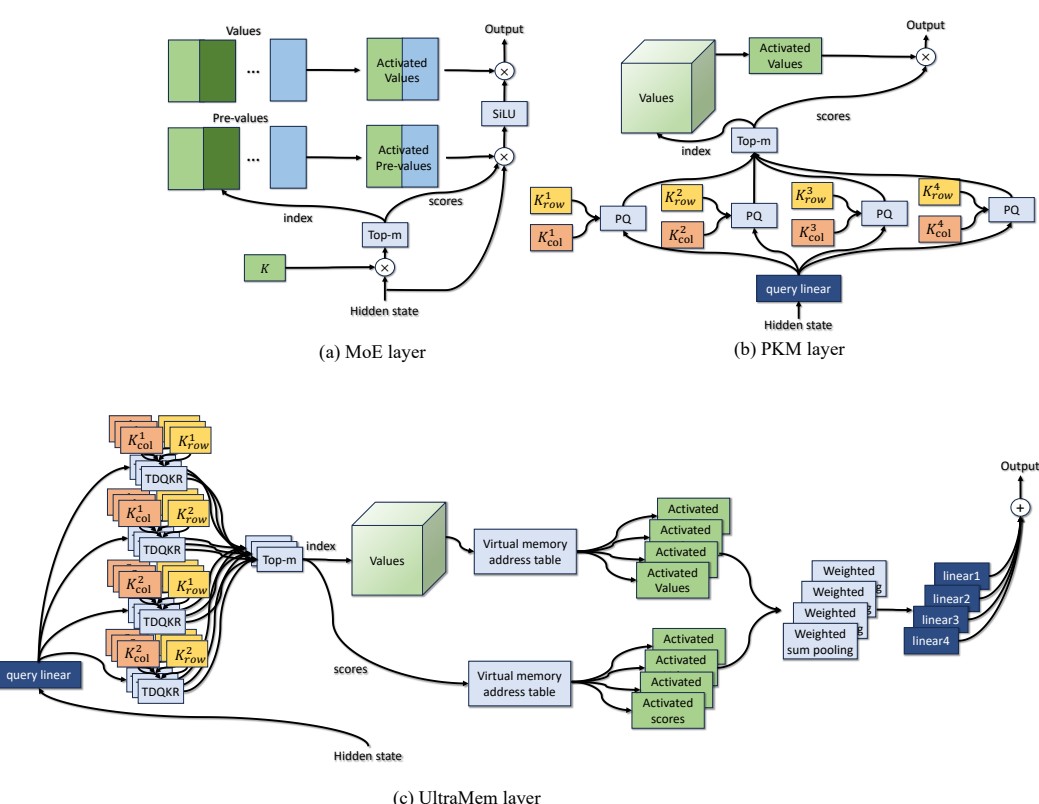

Figure 6: Overall structure of (a) MoE layer, (b) PKM layer, (c) UltraMem layer.

## B   OPTIMIZED INITIALIZATION

First, we list the important theorems required for the derivation, starting with the **Central Limit Theorem(CLT)**:

If $x_1, x_2, ..., x_n, ...$ are random independent samples of of a random variable $X$, then:

$$\frac{1}{\sqrt{n}} \sum_{i=1}^{n} (x_i - \mathbb{E}[X]) \to \mathcal{N}(0, \sigma(X)), \quad \text{as } n \to \infty \tag{16}$$

where $\sigma(X)$ is the standard deviation of the random variable $X$.

Notations:

- $\sigma_V$ is the standard deviation of the value in the memory layer.
- $h$ is the hidden size of the network.
- $d_{prev}$ is the dimension of the "Pre-values".
- $d_v$ is the dimension of the "Values"
- $k$ is the top-k parameter.
- $n_{head}$ is the head number.
- $L$ is the number of layers.
- $k_{inner}$ is the multiple obtained by dividing the FFN inner dimension by the hidden size.

- To control the final output magnitude, we adjust the initialization of the query and key norm such that the mean of the score obtained from "Top-k" is around 1, $\sigma_s$ is its standard deviation.

- The initialization standard deviation of the linears in attention and FFN is $\sqrt{\frac{2}{5h}}$ (Nguyen & Salazar, 2019). We also set the same standard deviation for the linear layers in UltraMemV2 for convenience.

Now, we can derive the output variance according to Figure 1.

## B.1   THE VARIANCE OF ULTRMEM-V2 LAYER OUTPUT

The input of the memory layer is the output of layer normalization before FFN. So the input to memory has a mean of 0 and a variance of 1. According to CLT, the output of "pre value proj" has a mean of 0 and a variance of 0.4. "Activated Pre values" have a mean of 0 and a variance of $\sigma_V^2$. Thus, the matrix multiplication result of these two tensors has a mean of 0 and a variance of $0.4 * \sigma_V^2 * d_{prev}$. This matrix multiplication result is then multiplied by the top-k score, this final score has a mean of 0 and a variance of $(0.4 + 0.4 * \sigma_s^2) * \sigma_V^2 * d_{prev}$.

Thus, the input of the "value proj" layer has a mean of 0, and a variance of $(0.4 + 0.4 * \sigma_s^2) * \sigma_V^4 * d_{prev} * k * n_{head}$. Finally, we can get the output variance using CLT:

$$\sigma_{mem}^2 = (0.16 + 0.16 * \sigma_s^2) * \sigma_V^4 * d_{prev} * k * n_{head} * d_v / h \qquad (17)$$

## B.2   THE VARIANCE OF TOP-K SCORE

There are many complicated operations in the "TDQKR", such as tucker and SVD, which makes it difficult for us to estimate the variance accurately. Therefore, we generate a large amount of random data whose distribution aligns with the output distributions of the query and key normalization processes. These data are then fed into the TDQKR module. Through statistical methods and tuning the initialization of the query and key norm, we ensure that the average score of the top-k results is approximately 1. Subsequently, we compute the variance of these scores under this condition and incorporate it into formula 17.

## B.3   CALCULATE THE STANDARD DEVIATION FOR INITIALIZATION

To control the variance of the final output, we take the initialized output variance of the FFN as a reference and directly set the output variance of each UltraMemV2 layer $\sigma_{mem}$ equal to that of the FFN $\sigma_{ffn}$, thereby deriving the initialized variance of the "Values" and "Pre-values" $\sigma_V$. Let's start by calculating the output variance of the FFN.

The input to the FFN is the output of layernorm or RMSNorm, and at the moment of initialization, its mean is 0 and variance is 1. Therefore, according to the Central Limit Theorem, the input to the Swish activation function has a mean of 0 and a variance of 0.4. Since the curve of the Swish activation function is quite similar to that of ReLU, a truncated normal distribution is used subsequently to estimate the distribution after Swish activation.

We use $\mu_{swi}$ and $\sigma_{swi}$ to denote the mean and std of the input to the Swish activation function; $a$ and $b$ are the boundary points of the truncate range, which are 0 and $+\infty$ respectively here. $\xi = \frac{x - \mu_{swi}}{\sigma_{swi}}$ convert the original normal distribution to the standard normal distribution. $\varphi(\xi) = \frac{1}{\sqrt{2\pi}} \exp\left(-\frac{1}{2}\xi^2\right); \alpha = \frac{a - \mu_{swi}}{\sigma_{swi}}; \beta = \frac{b - \mu_{swi}}{\sigma_{swi}}; \Phi(x) = \frac{1}{2}(1 + \text{erf}(x/\sqrt{2}))$, is the cumulative distribution function of the standard normal distribution; Let $Z = \Phi(\beta) - \Phi(\alpha)$; Then, the mean of the activated gate is:

$$\mu_{gate} = \mu_{swi} + \frac{\varphi(\alpha) - \varphi(\beta)}{Z} \sigma_{swi} \qquad (18)$$

The variance of the activated gate is:

$$\sigma_{gate}^2 = \sigma_{swi}^2 \left[ 1 - \frac{\beta\varphi(\beta) - \alpha\varphi(\alpha)}{Z} - \left( \frac{\varphi(\alpha) - \varphi(\beta)}{Z} \right)^2 \right] \qquad (19)$$

Then the activated gate is multiplied by another linear output whose mean is 0 and variance is 0.4 by using CLT. We substitute the specific values and obtain that the inner activation of FFN has a variance of 0.16, and a mean of 0.

To prevent the output variance of the final network from diverging as the number of network layers increases, the initialization standard deviation of the last linear layer in the FFN is usually further multiplied by a factor of $\sqrt{\frac{1}{2L}}$. Thus, by using CLT, the variance of the FFN output is:

$$\sigma^2_{ffn} = \frac{0.064 * k_{inner}}{2L} \tag{20}$$

We let $\sigma_{mem} = \sigma_{ffn}$ to get the initialzation variance:

$$\sigma^2_V = \sqrt{\frac{0.2 * k_{inner} * h}{k * n_{head} * (1 + \sigma_s^2) * d_{prev} * d_v * L}} \tag{21}$$

## C  EVALUATION BENCHMARK

For the proprietary model comparison, the evaluation benchmark including Open Benchmark, Hard Benchmark and Long-context Benchmark. Table 5 shows the components in Open Benchmark. Hard Benchmark contains more difficult tasks. Details of Long-context Benchmark is shown in Table 6. For the menchmark in open-source experiments, we evaluate models on Arc-C(Clark et al., 2018), Arc-E(Clark et al., 2018), CommonSenseQA(Talmor et al., 2019), Hellaswag(Zellers et al., 2019), MMLU-var(Muennighoff et al., 2024), OpenbookQA(Mihaylov et al., 2018), PIQA(Bisk et al., 2020), SCIQ(Johannes Welbl, 2017), and Winogrande(Sakaguchi et al., 2019).

Table 5: Open benchmarks across different domains

| Code | Math | Knowledge | Reasoning |
|---|---|---|---|
| MBPP(Austin et al., 2021) | Ape210K(Zhao et al., 2020) | MMLU(Hendrycks et al., 2021b;a) | Arc-C(Clark et al., 2018) |
| HumanEval(Chen et al., 2021) | GSM8K(Cobbe et al., 2021) | C-Eval(Huang et al., 2023) | BBH(Suzgun et al., 2022) |
| | MATH(Hendrycks et al., 2021c) | TriviaQA(Joshi et al., 2017) | DROP(Dua et al., 2019) |
| | | | WinoGrande(Sakaguchi et al., 2019) |
| | | | Hellaswag(Zellers et al., 2019) |

Table 6: Long-context evaluation tasks and their descriptions

| Task | Description |
|---|---|
| Long-context memo-rizing | Evaluate the model's ability to understand and recall information when there is a long context |
| Multi-round memo-rizing | Evaluate the model's ability to understand and recall information in the presence of multiple rounds of dialogue |
| In-context learning | Evaluate the model's capabilities under the given longer task demon-strations |
| Reasoning | Long context reasoning ability |
| Find needle | Evaluate the ability to quickly locate specific information in a large amount of information |
| Key-val retrieval | Given a large number of key-value pairs, evaluate the retrieval ability of the model when given keys |
| Multi-hop reasoning | Evaluate the model's ability to establish logical connections among dif-ferent context segments, thereby arriving at the answers to questions or making decisions |

## D    OPEN-SOURCE MODEL HYPERPARAMETERS

Table 7 details the configurations of our open-source models. We denote the number and dimension of keys as knum and kdim, respectively, while vdim and pre-vdim represent the dimensions of the value and pre-value. Notably, the Memory+-227M/1.2B share values across memory layers, resulting in a larger knum compared to UltraMem-227M/1.2B, although the number of values remains the same.

Table 7: Model Configurations

| Model | Layer | Hidden size | Attn head | Mem layer | Key number | $D_k$ | head × **TopM** | $D_v$ | $D_p$ | activated param(M) | total param(B) |
|---|---|---|---|---|---|---|---|---|---|---|---|
| OLMoE-227M/1.2B | 20 | 768 | 12 | / | / | / | / | / | / | 227 | 1.18 |
| Memory+-227M/1.2B | 20 | 768 | 12 | 4 | 1138 | 384 | 2x80 | 384 | / | 228 | 1.19 |
| UltraMem-227M/1.2B | 20 | 768 | 12 | 4 | 808 | 192 | 2x80 | 384 | / | 225 | 1.18 |
| UltraMemV2-227M/1.2B | 20 | 768 | 12 | 20 | 360 | 192 | 32 | 192 | 192 | 225 | 1.18 |
| OLMoE-1B/7B | 16 | 2048 | 16 | / | / | / | / | / | / | 1070 | 6.71 |
| UltraMemV2-1M/7B | 16 | 2048 | 16 | 16 | 528 | 512 | 128 | 768 | 384 | 1079 | 6.70 |

Table 8: Value dimension and pre-value dimension Ablation Configurations

| Model | Hidden size | Mem layer | Key number | $D_k$ | head x **TopM** | $D_v$ | $D_p$ |
|---|---|---|---|---|---|---|---|
| $D_p : D_v = 1 : 3$ | 1152 | 24 | 964 | 524 | 94 | 216 | 72 |
| $D_p : D_v = 1 : 8$ | 1152 | 24 | 964 | 524 | 94 | 256 | 32 |

## E    OTHER ABLATIONS

### E.1    SINGLE PROJECTOR AND MULTI QUERIES

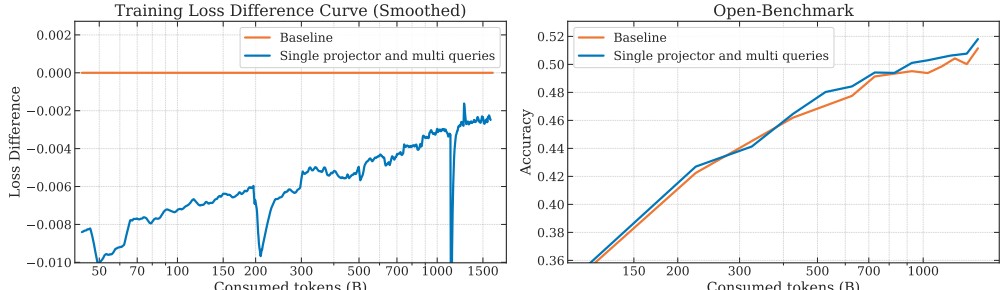

Figure 7: Single projector and multi-query modifications on UltraMemV2-430M/5B. Training loss difference (left) and Open-Benchmark accuracy (right) comparing baseline with the proposed approach. The modifications achieve 0.0026 loss reduction and 0.7-point accuracy improvement after 1.5T tokens.

Under the condition of maintaining the same parameter count and computational load, allocating more activations to the final value projector requires reducing activations in the FFN. Our research reveals that value projectors are not highly activation-efficient; thus, we use only a single projector for the final derived value. Additionally, we employ separate queries for each tucker rank, which enhances the accuracy of query-key operation results. After training 1.5T tokens, these two modifications yield a loss reduction of 0.0026 and a 0.7-point improvement in Open-Benchmark performance as shown in Figure 7.

### E.2    1 HEAD

We also performed an ablation study on the number of heads in UltraMemV2-430M/5B. For 1 head configuration, we double the $D_k$ and TopM to maintain the same computation and value retrieval. As

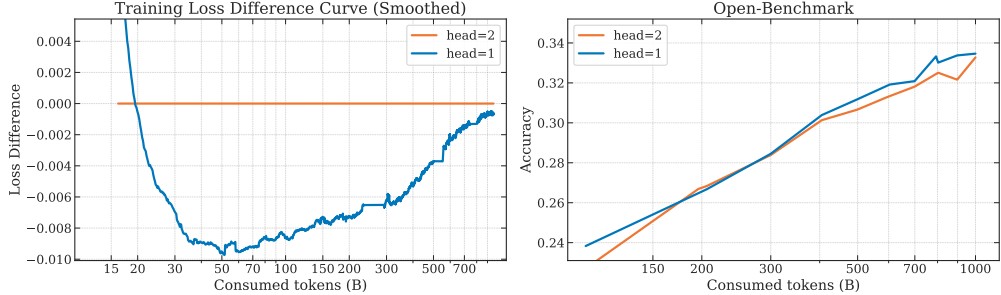

Figure 8: Ablation study on the number of heads in UltraMemV2-430M/5B. Training loss difference (left) and Open-Benchmark accuracy (right) comparing single head vs. two heads. Using a single head achieves 8e-4 loss reduction and 0.2-point accuracy improvement after 1T tokens.

illustrated in Figure 8, after training with 1T tokens, using a single head yields a marginal improvement over using two heads: the loss decreases by 8e-4 and the Open Benchmark score increases by 0.2 points. This phenomenon can be attributed to the fact that while employing a single head increases the number of values to be retrieved, the dimensionality of each query and key also increases proportionally. Ultimately, the gain in retrieval accuracy from the increased dimensionality of the retrieval vector marginally outweighs the negative impact of the larger candidate pool for retrieval.

### E.3 THE COMPUTATIONAL PROPORTION OF ULTRAMEMV2

The larger the keys dimension $D_k$, the greater the computational proportion of memory. To keep the total amount of computation unchanged, the computation of FFN will decrease. Therefore, if $D_k$ is too large, the computing power allocated to FFN will be too small, which will lead to poor performance; if $D_k$ is too small, the query result in the memory layer will be very inaccurate, which will also result in poor performance.

To determine how to configure $D_k$, we conduct detailed ablation studies on UltraMemV2-500M/6B. We only adjusted $D_k$ and correspondingly tweaked the inner dimension of the FFN to ensure that the total computational cost and the total number of parameters remained consistent. The UltraMemV2 related configurations are as follows: (MCP is short for Memory Computational Proportion in Table 9)

Table 9: Key dimension Ablation Configurations

| Model | Hidden size | Mem layer | Key number | $D_k$ | head x TopM | $D_v$ | $D_p$ |
|---|---|---|---|---|---|---|---|
| MCP=12.0% | 1152 | 24 | 964 | 344 | 94 | 144 | 144 |
| MCP=14.5% | 1152 | 24 | 964 | 432 | 94 | 144 | 144 |
| MCP=17.0% | 1152 | 24 | 964 | 524 | 94 | 144 | 144 |
| MCP=19.5% | 1152 | 24 | 964 | 610 | 94 | 144 | 144 |
| MCP=23.0% | 1152 | 24 | 964 | 730 | 94 | 144 | 144 |
| MCP=25.0% | 1152 | 24 | 964 | 796 | 94 | 144 | 144 |
| MCP=27.5% | 1152 | 24 | 964 | 880 | 94 | 144 | 144 |

As shown in Figure 9, the loss performance shows that 17% Memory Computational Proportion is the best configuration. When scaling up the model, if the hidden size increases by a factor of $\alpha$, both vdim and pre-vdim will also increase by a factor of $\alpha$. This causes knum to increase only by a factor of $\sqrt{\alpha}$, and the proportion of computational load for memory will decrease as the model grows larger. Therefore, it is not suitable as a basis for scaling up the model. Finally, when we scale up the model, we adopt $D_k = h/2$ as a reasonable configuration.

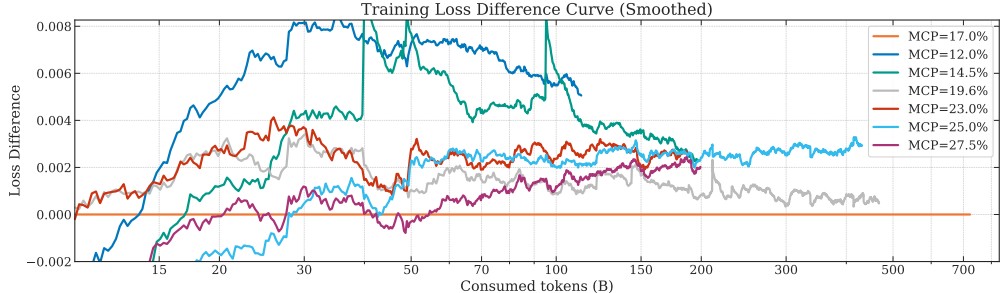

Figure 9: Training loss difference under different memory computational proportion.

### E.4 AUXILARY LOSSES

In this section, we introduce two auxiliary losses which are **NOT** used in UltraMemV2. Our experements show no improvement under these auxiliary losses.

**Tucker core penalty loss** In TDQKR, when doing the first two `TopM`, Huang et al. (Huang et al., 2024) first do the Singular Value Decomposition of the tucker core and aggregate the row and column keys with the eigenvectors of the largest eigenvalues. To constrain this approximation error, they place constraints on non-maximum singular values

$$\mathbf{C} = \mathbf{U}\Lambda\mathbf{T}^\top, \quad \text{(by SVD)} \tag{22}$$

$$\mathcal{L}_{aux} = \frac{\alpha}{r-1}\sum_{i=2}^{r}\left(\max\left(0, \lambda_i - \tau\right)\right)^2, \tag{23}$$

where, $\Lambda$ denotes the singular values for $\mathbf{C}$ in descending order, with $\tau$ serving as a margin to prevent $\mathbf{C}$ from degenerating into a rank-1 matrix, and $\alpha$ is the coefficient for the loss.

**Balance loss** in MoE can solve the problem of dead experts and alleviate unbalanced computation of Expert Parallel(He et al., 2021). Due to the fact that UltraMem's parallelism is segmented in the embedding dimension, there is no problem of computational imbalance. We are curious whether a more balanced activation embedding will also improve the performance. Recall that the balance loss of MoE(Xie et al., 2023) is a constraint on the result of the router, while UltraMem can be thought of as a MoE with row/column routers, so we propose the balance loss of UltraMem following MoE. Let $\mathbf{u}, \mathbf{t} \in \mathbb{R}^{r \times 1}$ be the eigenvectors corresponding to the largest eigenvalues, the probability to activate each row/column key is

$$\mathbf{P_{row}} = Softmax(\mathbf{u}^\top\mathbf{S}_{row}), \quad \mathbf{P_{col}} = Softmax(\mathbf{v}^\top\mathbf{S}_{col}), \tag{24}$$

here we omit the head index $i$ for brevity. Ones we get probabilities, we can calculate the balance loss

$$\mathcal{L}_{balance} = \beta N \cdot \sum_{n=1}^{N} f_n \cdot p_n, \tag{25}$$

where $\beta$ is the coefficient for the loss, $N$ is the number of row/column keys, $f_n$ represents the frequency at which row/column keys is activated. Given a batch B with T tokens, $Count_n$ is the number of times the $n$-th row/column key is activated, then

$$f_n = Count_n/T, \quad p_n = \frac{1}{T}\sum_{t}^{T}\mathbf{P_{row/col}^n}, \tag{26}$$

where $p_n$ represents the average probability of the $n$-th row/column key being selected.

### E.4.1 EXPERIMENTS

In this subsection, we conduct experiments on UltraMemV2-430M/5B, training 800B tokens.

**Tucker core penalty loss** In contrast to the approach in UltraMem (Huang et al., 2024), which emphasizes the necessity of constraining non-maximum eigenvalues of the Tucker core to mitigate approximation errors, our empirical findings suggest that such a constraint may be superfluous during training. We observe that the eigenspectrum of the Tucker core exhibits a sharp decay, where the principal eigenvalue, $\lambda_1$, is naturally an order of magnitude larger than the subsequent eigenvalues. As illustrated in Figure 10.b, $\lambda_1$ is consistently more than four times larger than the second-largest eigenvalue, $\lambda_2$. Given this substantial gap, the resulting approximation error from the non-maximum eigenvalues is negligible.

To further validate this, we conducte an ablation study by removing the Tucker core penalty loss proposed in UltraMem. Figure 10.a shows the downstream task performance throughout training. The model trained without the penalty loss maintains comparable accuracy to the baseline in the early training stages. Notably, after training on 800B tokens, our model without the penalty exhibits a 0.3 point improvement in accuracy, indicating that forgoing the explicit eigenvalue constraint does not compromise, and may even slightly benefit, model performance.

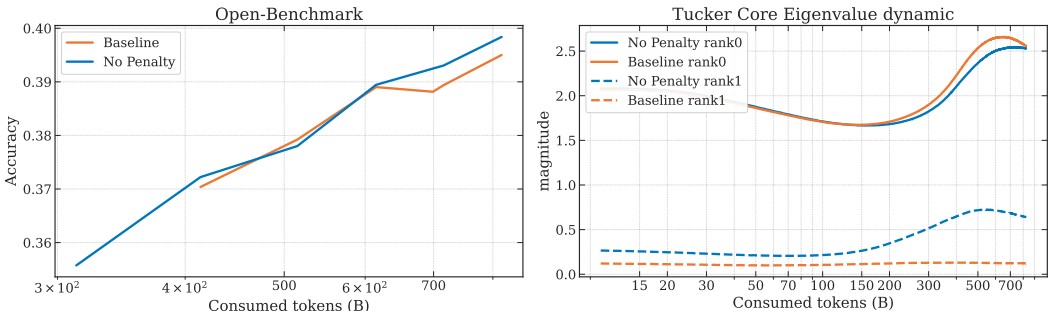

Figure 10: Tucker core penalty loss ablation study. Left: Open-Benchmark accuracy during training with and without Tucker core penalty loss. Right: Evolution of Tucker core eigenvalue magnitudes, showing the dominant eigenvalue $\lambda_1$ (rank0) and second-largest eigenvalue $\lambda_2$ (rank1). The penalty loss maintains eigenvalue separation without compromising downstream performance, validating that explicit eigenvalue constraints are unnecessary when the natural eigenvalue gap is sufficient.

**Balance loss** is a common technique in MoE to promote load balancing across experts. As demonstrated in OLMoE (Muennighoff et al., 2024), it can lead to improved training loss and downstream task accuracy. Typically, balanced expert utilization is also critical for training efficiency. However, in the UltraMemV2, parallelism is implemented along the value dimension, which renders training speed independent of the value selection balance. We are nonetheless interested in the impact of the balance loss on model performance. To this end, we conduct experiments on UltraMemV2-430M/5B, incorporating the balance loss from Equation 25 with a coefficient of $\beta = 0.001$.

As illustrated in Figure 11, our findings reveal a dependency on the number of activated values (`TopM`). When a smaller number of values are activated (`TopM` $= 47$ out of value number $N = 465124$), the inclusion of the balance loss correlated with a clear improvement in both training loss and downstream accuracy. Conversely, when a larger number of values are activated (`TopM` $= 94$), the application of the balance loss has a detrimental effect on performance, with both training loss and downstream accuracy degrading. These conflicting results indicate that the effectiveness of the balance loss is linked to the number of activated values. The loss provides a regularizing benefit when this number is small, but this benefit is lost when the number of activated values becomes too large.

### E.5 SHARED MEMORY

Inspired by shared memory models like PKM (Lample et al., 2019) and Memory+ (Berges et al., 2024), which utilize a shared value table across multiple layers, we explore similar sharing paradigms. Unlike MoE where inter-layer expert sharing significantly inflates inference memory access for small batches, memory access in memory layer remains constant. However, TDQKR inherently increases index computation: it doubles with rank $r = 2$, and again doubles for each layer if, for instance, one value table is shared across four layers. To maintain computational parity,

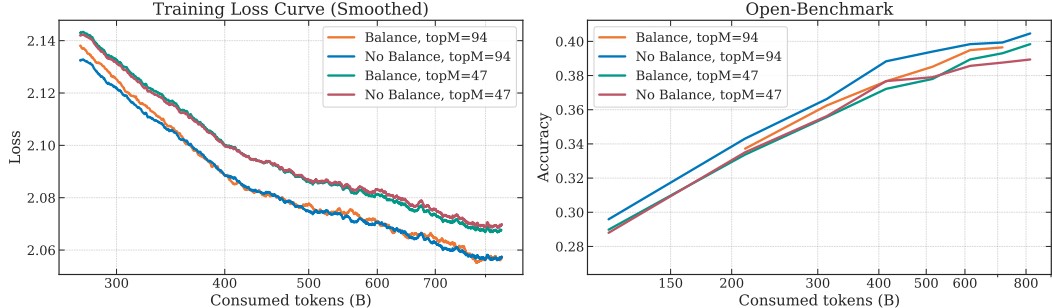

Figure 11: Effect of balance loss on training dynamics with different numbers of activated values. Training loss (left) and downstream accuracy on Open-Benchmark (right) for UltraMemV2-430M/5B with and without balance loss ($\beta = 0.001$). Balance loss improves performance when fewer values are activated (TopM = 47) but degrades performance with more activated values (TopM = 94), indicating that the regularization benefit depends on the sparsity level.

this necessitates shrinking the FFN inner dimension, potentially degrading model performance due to reduced capacity. To address this, our experiments evaluate the efficacy of partially sharing the value table.

**Experimental Setup** Our experiments utilize the UltraMemV2-500M/6.8B model as a baseline, configured with $L = 24$ transformer layers, TopM = 94 activated values, row/column key number $n = 964$, and $N = 9292296$ values per table. We define three primary sharing strategies:

1. S$g$-NoRing: Each memory layer accesses its own value table plus those of its $g-1$ nearest neighboring layers (e.g., $(g-1)/2$ preceding and $(g-1)/2$ succeeding, adjusted for boundaries where $g$ is odd; if $g$ is even, it might be $g/2$ preceding and $(g/2)-1$ succeeding, or a similar asymmetric but consistent distribution).

2. S$g$-Ring: Similar to S$g$-NoRing, but with wrap-around, allowing layers near the beginning/end of the network to access tables from the end/beginning, respectively, ensuring each layer can access $g$ distinct tables.

3. S$g$-Block: Layers are grouped into contiguous blocks of $g$; layers within a block share all $g$ value tables belonging to that block, but do not access tables outside their block.

Table 10 provides an illustrative example of these schemes. As the number of shared tables $g$ increases, the effective key dimension for lookup also increases (e.g., for sharing 4 value tables, the composite key number $n$ becomes 1928). We correspondingly reduce the FFN inner dimension to ensure iso-computation across all configurations.

Table 10: Example of different shared value partern

| Layer number | $L = 1$ | $L = 2$ | $L = 3$ | $L = 4$ | $L = 5$ | $L = 24$ |
|---|---|---|---|---|---|---|
| S4-NoRing | 1,2,3,4 | 1,2,3,4 | 2,3,4,5 | 3,4,5,6 | 4,5,6,7 | 21,22,23,24 |
| S4-Ring | 24,1,2,3 | 1,2,3,4 | 2,3,4,5 | 3,4,5,6 | 4,5,6,7 | 23,24,1,2 |
| S4-Block | 1,2,3,4 | 1,2,3,4 | 1,2,3,4 | 1,2,3,4 | 5,6,7,8 | 21,22,23,24 |

**Ring vs. NoRing Topology** We first ablate the S9-NoRing and S9-Ring configurations. As depicted in Figure 12, S9-NoRing achieves a lower training loss but exhibits a 1-point degradation in downstream accuracy compared to S9-Ring. This suggests S9-NoRing may be more susceptible to overfitting. We hypothesize this could be due to the values in tables at the network's extremities being more frequently shared and thus potentially over-specializing.

**Optimal Number of Shared Layers** We then investigate the impact of the number of shared layers $g$ within the Ring topology, specifically comparing S4-Ring, S9-Ring, and S16-Ring. These correspond to effective key number $n$ of 1928, 2896, and 3856, respectively. Figure 13 shows that increasing $g$ provides a continuous benefit to training loss relative to the baseline. However, this

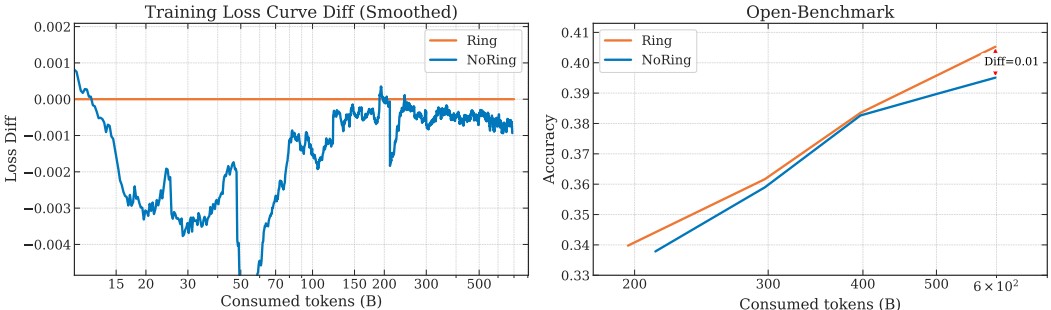

Figure 12: Comparison of Ring vs. NoRing topologies for shared memory configurations. Left: Training loss difference (smoothed) showing NoRing topology achieves lower training loss. Right: Open benchmark accuracy demonstrating the advantages of Ring, which achieving better final accuracy (Diff=0.014).

benefit appears to saturate at $g = 9$, with S16-Ring showing negligible improvement over S9-Ring in training loss. In terms of downstream accuracy, S9-Ring surpasses the baseline by 1 point, while S16-Ring does not improve upon S9-Ring. This indicates a trade-off: while access to a larger pool of shared values can enhance representational capacity, the requisite FFN shrinkage to maintain computational parity eventually negates further gains.

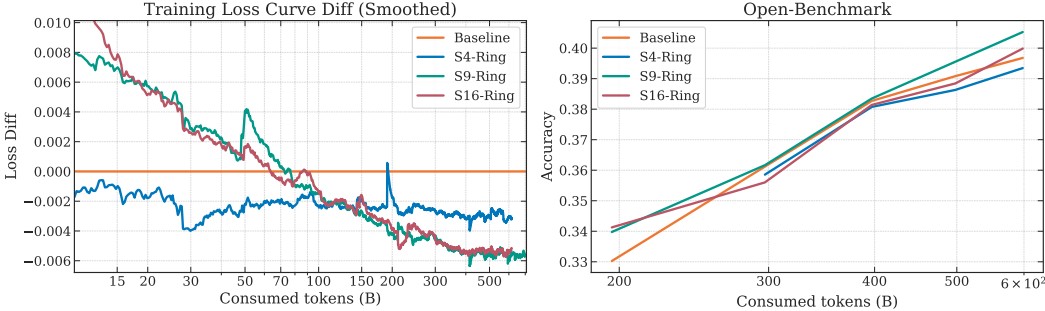

Figure 13: Impact of the number of shared layers in Ring topology configurations. Left: Training loss difference (smoothed) relative to baseline showing S4-Ring, S9-Ring, and S16-Ring all achieve lower training loss, with benefits saturating around 9 shared layers. Right: Open benchmark accuracy demonstrating S9-Ring achieves the best downstream performance with 1-point improvement over baseline, while S16-Ring shows no improvement over S9-Ring.

**Block-wise Sharing for Scalability** Finally, considering the practicalities of large-scale model training, particularly with pipeline parallelism, Ring-based sharing can introduce significant inter-stage communication overhead. Block-wise sharing (S$g$-Block) presents a more engineering-friendly alternative. We evaluate S6-Block, as with $L = 24$ layers, this creates 4 blocks of 6 layers, offering a comparable degree of value table accessibility to S9-Ring (where each layer accesses 9 tables, but tables are re-used across layers). Figure 14 compares the baseline, S4-Ring, S9-Ring, and S6-Block. Results indicate that S6-Block, while marginally inferior to S9-Ring, still offers substantial improvements over the baseline. It is expected that within a larger share range, the effect of S$g$-Block can be further enhanced. This validates S$g$-Block as a promising strategy for large model training, balancing performance with practical implementation constraints.

### E.6 VALUE LEARNING RATE SCHEDULE

The UltraMem(Huang et al., 2024) demonstrates efficacy by employing a relatively high initial learning rate for its value parameters, which subsequently decayed. This strategy, while effective, introduces two additional hyperparameters: the initial learning rate multiplier and the decay duration. To investigate the necessity of this decaying schedule and potentially simplify the training

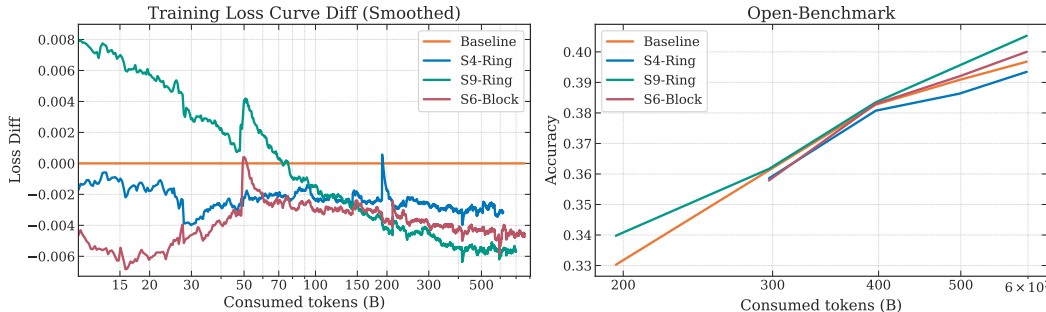

Figure 14: Comparison of block-wise sharing topology (S6-Block) against ring-based topology. Left: Training loss difference (smoothed) showing S6-Block achieves comparable training loss improvements to S9-Ring while being marginally inferior. Right: Open benchmark accuracy demonstrating S6-Block offers substantial improvements over baseline with performance Slightly worse than S9-Ring, validating block-wise sharing as a practical alternative for large-scale model training.

regimen, we conduct an ablation study on UltraMemV2-500M/6.8B. We compare three settings for the pre-value and value learning rates, all trained for 1.4T tokens:

1. Baseline: An initial rate of 4x the main model's learning rate, linearly decaying to 1x by 350B tokens, mirroring the UltraMem approach.

2. Constant 1x: A constant rate of 1x the main model's learning rate.

3. Constant 1.5x: A constant rate of 1.5x the main model's learning rate.

Figure 15 depicts the training loss and downstream task accuracy for these configurations. The baseline initially achieves the lowest training loss, peaking in its advantage around 430B tokens. However, this gap progressively narrows, and by 1.4T tokens, the loss differences across all settings become negligible. A consonant trend is observed in downstream accuracy: the baseline shows its most significant lead around 400B tokens. Notably, upon completion of 1.4T tokens of training, the constant 1x learning rate setting surpasses the baseline by 0.4 points in downstream accuracy. The constant 1.5x setting also demonstrates superior performance to the 1x setting prior to 300B tokens, after which its relative performance deteriorates.

These findings suggest that while a higher, decaying learning rate for the pre-value and value parameters provides an early advantage in training, particularly for shorter training budgets (fewer tokens), this benefit diminishes with extended training. In fact, for sufficiently long training horizons, maintaining a constant, moderate learning rate (e.g., 1x the main model's learning rate) can yield superior final performance, potentially obviating the need for a decaying schedule and its associated hyperparameter tuning.

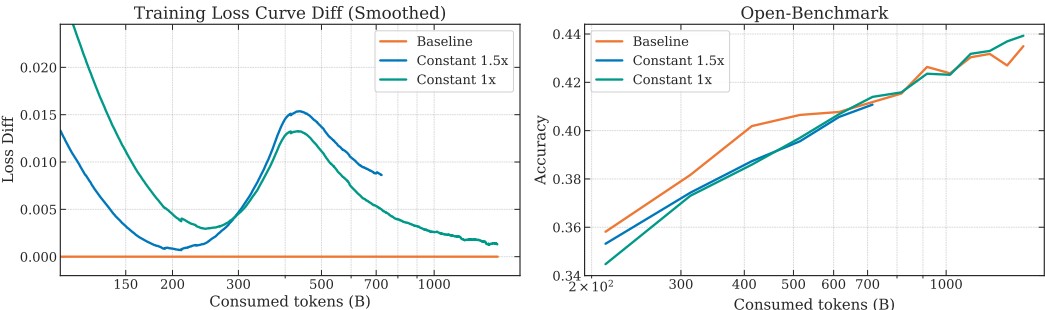

Figure 15: Impact of value learning rate schedules on training dynamics and downstream performance. Left: Training loss difference (smoothed) showing the baseline (4x→1x decay) initially achieves lowest loss but converges with constant rate settings by 1.4T tokens. Right: Open-domain benchmark accuracy demonstrating constant 1x learning rate achieves best final performance (0.4 point improvement), suggesting decaying schedules provide early training benefits that diminish over extended training horizons.

