# OpenReview forum: "UltraMemV2: Memory Networks Scaling to 120B Parameters with Superior Long-Context Learning"
_ICLR.cc/2026/Conference — ICLR 2026 Poster_

### Official Review · Reviewer_rcay · 2025-10-26

**Soundness:** 3
**Presentation:** 3
**Contribution:** 3
**Rating:** 6
**Confidence:** 3

**Summary:**

UltraMemV2 refines the Memory Layer architecture as a new sparse alternative to MoE models.
While the original UltraMem only reached the performance level of a 2 active expert MoE, UltraMemV2 achieves performance comparable to an 8 active expert MoE, marking a significant advancement.
The authors introduce several key improvements, including a memory layer in each Transformer block, a simplified implicit value expansion (IVE), a PEER-based feedforward mechanism, improved initialization for stable training, and optimized compute ratios between memory to  FFN.
With these enhancements, UltraMemV2 shows notable gains (+6 to +8 points) on long-context, multi-step reasoning, and memory-intensive tasks, while efficient memory access comparable to existing MoE models.

**Strengths:**

1. Considering that MoE architectures have become a de facto standard component in LLM training, the direction of this work is highly meaningful. Improving the cost efficiency and performance of such sparse architectures is an important and timely.

2. The reported performance is very promising, matching the top-k=8 MoE configuration that is commonly used in recent LLM models.

3. The experiments are conducted at a large scale up to 120B parameters with 2.5B active and trained on 4.4T tokens which makes the results convincing and demonstrates the method’s scalability.

**Weaknesses:**

1. It would be valuable to include more discussion or quantitative results about GPU hours, training/inference latency, and throughput, especially compared to existing MoE models.
These metrics would strengthen the claim that UltraMemV2 offers practical efficiency gains.

2. While hyperparameter search is generally required for LLM training, the proposed model appears to be more sensitive to hyperparameter choices, such as initialization and learning-rate scheduling.

**Questions:**

1. It is mentioned that UltraMemV2 cannot easily achieve higher sparsity. Did the authors analyze whether its sparsity scaling behavior differs from that of MoE models?

---

> ### Author Response · Authors · 2025-11-19
> **Authors' Response**
>
> 1. We completely agree with this suggestion. We have added a comparison of inference speed between MoE and UltraMemV2 (both with 10B active parameters and 200B total parameters) in the revised paper. The results show that UltraMemV2 achieves over 1.5× faster inference speed compared to MoE. We believe these quantitative metrics strengthen our efficiency claims and appreciate you pointing out this important addition.
> 2. Thank you for raising this concern. We'd like to clarify that while UltraMemV2's sparsity does require careful training, we have largely addressed stability issues through query and key normalization, as shown in Figure 10(a) of the paper[1].
>     1. Regarding initialization, our approach is derived from standard FFN initialization and represents a natural generalization rather than a specialized scheme designed specifically for UltraMemV2.
>     2. For learning rate scheduling, we describe experiments with 1× and 4× learning rates for the value in Appendix E6. Interestingly, we found that the higher learning rate strategy proposed in the original paper[1] can actually be removed when training with sufficient data (1T tokens), which simplifies the training process.
>   Overall, we believe UltraMemV2's training is not particularly hyperparameter-sensitive once these normalization techniques are applied.
> 3. We'd like to clarify our statement: when improving UltraMemV2's performance, reducing sparsity is more effective than increasing total parameters, though this comes with slower inference speed. In contrast, increasing parameters has minimal impact on UltraMemV2's inference speed. It's worth noting that our UltraMemV2 model with 2.5B active parameters already operates at 64× the sparsity of comparable MoE models, which is a very high sparsity level. Regarding sparsity scaling behavior, Figure 6(b) in the original UltraMem paper [1] shows how sparsity affects loss, and the trend appears similar to MoE models. We believe UltraMemV2 and MoE exhibit comparable sparsity scaling properties, though a more systematic analysis would be valuable future work.
> [1] https://arxiv.org/pdf/2411.12364

---

> > ### Comment · Reviewer_rcay · 2025-11-19
> >
> > Thank you for the detailed response, most of my concerns are resolved. The reported improvement in inference latency is impressive.
> >
> > However, it appears that the throughput metrics for both training (pretraining) and inference are not included in the updated version. While I acknowledge that throughput figures depend on the software stack and implementation details, I believe providing such comparisons or discussions would still be beneficial for the readers. Since there are scenarios where throughput is more critical (e.g., pretraining under a limited compute budget or rollouts with large batch sizes in RL), including these perspectives would further strengthen the paper.

---

> > > ### Author Response · Authors · 2025-11-20
> > > **Authors' Response**
> > >
> > > Thank you for your suggestion. It was our oversight.
> > >
> > > We have now added detailed training throughput analysis and comparisons in Section 4.3 of the revised paper. Our findings show that UltraMemV2 and SeedMoE achieve essentially equivalent training throughput. This parity is expected and easy to understand: during training with large batch sizes, both architectures activate all parameters, eliminating the memory access advantages that UltraMemV2 enjoys during inference. In contrast, inference throughput improves by 1.5–2× for UltraMemV2, thanks to its extreme sparsity, which significantly reduces memory bandwidth requirements.
> > >
> > > We hope these clarifications strengthen the paper's practical relevance, and we hope the revisions will prompt reconsideration of our score.

---

> > > > ### Comment · Reviewer_rcay · 2025-11-20
> > > >
> > > > Thank you for the update. All my concerns have been resolved, and I have raised my score.

---

### Official Review · Reviewer_f3Vh · 2025-10-31

**Soundness:** 4
**Presentation:** 3
**Contribution:** 2
**Rating:** 6
**Confidence:** 3

**Summary:**

The paper proposed a sparse memory architecture by tweaking memory layer architecture to match the performance with MOEs with 8 experts. The paper does a good job of providing a Comprehensive analysis and detailed ablation studies. The paper also discusses scalability, though it’s unclear if the same scaling law of LLMs holds with trainable memory parameters.
It improves inference efficiency over prior work, by  simplifying value expansion with single linear projections, demonstrating that parameter efficiency of non-shared linear layers is actually not high. This paper also overcomes the limitation of number of memory layers, where previous work has shown degradation of performance if the number of memory layers are too high.

**Strengths:**

1. Good ablation for number of layers and  overcomes the limitation of number of memory layers
2. Matched performence with MOEs with 8 experts
3. Uses strong benchmarks and evaluation
4. Simplifies the  value expansion, making inference more efficient

**Weaknesses:**

Overall the contribution is light, the paper aims to bridge the gap(in performance) between MOEs and Memory layer architectures. In terms of scientific novelty, some of the approaches seem incremental and this approach seems to combine multiple incremental tweaks to achieve performance improvements over baseline. For example, “Memory Layer at Scale”(Berges, 2024) paper demonstrated that  was multiple memory layers increase performance significantly over having a single layer(In their case, performance degraded going beyond 3 layers).  Another example is adoption of simplified value expansion with a small tweak of single linear projection.

**Questions:**

1. A more rigorous scaling law analysis and discussion around scaling trends are required if this is proposed as an alternative architecture to MoEs
2. Can this approach be introduce during mid-training or post training and achieve good performance. Any discussion around the performance gap with CT would be great.
3. There are other parametric memory work, such as memory Layers at scale. It would be good to compare the results with such alternative approaches.
4. It would be good to share results on how high quality RAG combines with this approach and how does this approach compares with RAG for long context tasks

Minor comments:
1. Discussion about how can this be combined with MOEs

---

> ### Author Response · Authors · 2025-11-19
> **Authors' Response**
>
> 1. We agree that a rigorous scaling law analysis would be valuable. Our goal was to first establish the model architecture and training recipe before extensively exploring scaling properties. From our current experiments and the preliminary scaling law analysis in the original UltraMem paper, UltraMemV2 demonstrates scaling properties similar to MoE models. We will pursue a comprehensive scaling law study as future work.
> 2. Yes, memory layers can be straightforward to add during mid-training. However, converting a pretrained MoE to UltraMemV2 is more challenging. A simple approach would be to insert UltraMemV2 layers into an existing MoE model and initialize the value projection with zeros, or alternatively use a gating scalar initialized to zero. This allows the memory layers to be gradually incorporated without disrupting the pretrained model's behavior. Recent study (Continual Learning via Sparse Memory Finetuning, https://arxiv.org/abs/2510.15103) has found that integrating the memory layer during the fine-tuning phase can alleviate catastrophic forgetting, and we believe that the integration of UltraMemV2 during the CT and fine-tuning phases has extensive research value. We would love to add a discussion section to our paper.
> 3. Thank you for this suggestion. We have actually included a comparison with Memory+ (from "Memory Layers at Scale") in Table 3 of our paper. The results show that UltraMemV2 outperforms Memory+ on both training/eval loss and downstream benchmarks. We appreciate you highlighting this important baseline, and we're happy to expand the discussion of this comparison in our revision if it would be helpful.
> 4. Thank you for this interesting suggestion. We haven't explored combining UltraMemV2 with RAG yet, but we agree it's a promising direction—the efficient routing mechanism in UltraMemV2 could potentially leverage external information from RAG effectively. We'd like to clarify that UltraMemV2 demonstrates advantages over MoE on long context tasks as standalone architecture. Since both MoE and UltraMemV2 could be augmented with RAG, a fair comparison would be MoE+RAG vs. UltraMemV2+RAG on long context benchmarks. We believe this is an exciting avenue for future work and plan to explore it in follow-up studies.
> 5. To Minor comments: Thank you for this suggestion. We have added a discussion section addressing how UltraMemV2 can be combined with MoE architectures. The key insight is that these approaches are complementary rather than mutually exclusive—UltraMemV2 layers can be integrated alongside MoE layers within a hybrid architecture, or introduced into existing models during mid-training using zero initialization. We discuss the practical approaches and potential benefits of such combinations in the revised paper.

---

### Official Review · Reviewer_hfVp · 2025-11-01

**Soundness:** 3
**Presentation:** 3
**Contribution:** 4
**Rating:** 8
**Confidence:** 3

**Summary:**

The paper presents UltraMemV2, a memory-layer architecture intended to close the performance gap between memory-layer sparse approaches and Mixture-of-Experts models. The authors introduce five main changes: placing memory layers in every transformer block, simplifying implicit value expansion to a single linear projection, using FFN-based value processing inspired by PEER, a new initialization for the memory layer, and rebalancing memory and FFN computation proportions. They evaluate UltraMemV2 across proprietary and open benchmarks and report that UltraMemV2 reaches parity with 8-expert MoE under matched compute/parameters while requiring much lower memory access. They highlight particularly strong gains on memory-heavy tasks and validate scaling up 120B parameter models.

**Strengths:**

Strong architectural contributions: the five design changes proposed by the authors are all justified through ablations and contribute to improved model performance

Strong empirical evaluation: multiple model scales up to 120B and a diverse selection of benchmarks make the authors claims very convincing.

Initialization analysis: the paper contributes a new initialization scheme to stabilize training of the memory layer, which addresses a common failure mode for large sparse modules.

Practical use: UltraMemV2 is a compelling architecture for deploying models under memory bandwidth constraints because of the relatively lower memory accesses.

**Weaknesses:**

The paper motivates UltraMemV2 with lower memory access and inference cost, but it would be more convincing to see latency and bandwidth comparisons vs. traditional MoE models

Proprietary data: this might be unavoidable but the proprietary nature of the benchmarks and data limits the reproducibility of the methods in this paper

The UltraMemV2 model has significantly worse benchmark performance on multi-hop reasoning. The paper would be improved if the authors investigated this further and demonstrated through other benchmarks whether UltraMemV2 has worse overall reasoning abilities or if it is specific to this benchmark.

**Questions:**

The method lags behind early in training as mentioned by the authors. Do you have more details and possible intuitions why UltraMemV2 is slower to train initially? And what does "early" mean in general - how much data does it need to catch up?

Is there a reason for the drop in multi hop reasoning? Is this reflective of the model's overall reasoning capabilities on downstream tasks?

Could you quantify the inference cost improvements that you alluded to eg the reduction in memory access?

---

> ### Author Response · Authors · 2025-11-19
> **Authors' Response**
>
> Thank you for these thoughtful comments and questions. We address each point below:
> 1. To Weakness-1 and Question-3: We completely agree. We have now added detailed comparisons of memory access and inference costs in the revised paper. Please take a look at the new results. For models with 10B active parameters and 200B total parameters, UltraMemV2 achieves at least 1.5× faster inference compared to MoE.
> 2. To Weakness-2: We understand this concern and have taken steps to maximize reproducibility. We have open-sourced our training code, and half of our evaluation uses open benchmarks (with detailed breakdowns provided in the Appendix). We believe these efforts make our methods as reproducible as possible given practical constraints.
> 3. To Weakness-3 and Question-2: We agree this requires further investigation. Importantly, UltraMemV2 performs on par with MoE on standard reasoning benchmarks (Table 1, both OpenBench and HardBench), suggesting the issue is specific to multi-hop reasoning rather than reasoning ability in general. We suspect the extreme sparsity may play a role, but this requires further experimental validation. We are optimistic that supervised fine-tuning (SFT) and reinforcement learning (RL) stages may address this gap.
> 4. To Question-1: We hypothesize that the primary reason is extreme sparsity. UltraMemV2-2.5B/60B-top768 has 64× the sparsity of SeedMoE-2.5B/60B, which naturally leads to slower convergence. Additionally, we observe that when memory layers achieve the same loss as MoE, they show lower loss on easy-to-predict tokens but higher loss on hard-to-predict tokens (Here, the lower and higher levels are very slight, but they can be observed), suggesting greater susceptibility to overfitting. Since pretraining (PT) data quality is lower than continued training (CT) data quality, this explains why UltraMemV2 underperforms MoE during PT but surpasses it after CT. By "early," we refer to the lower-quality data phase in pretraining, typically followed by high-quality CT. Our experiments show that 250B tokens of high-quality data are sufficient for UltraMemV2 to catch up to and surpass MoE.

---

### Official Review · Reviewer_3bZy · 2025-11-03

**Soundness:** 3
**Presentation:** 3
**Contribution:** 2
**Rating:** 6
**Confidence:** 3

**Summary:**

This paper provides a memory‑layer architecture intended as an alternative to MoEs, which is an extension of UltraMem. The novelty with respect to the origianl UltraMem are, adding a memory layer to every Transformer block, simplifying the implicit value expansion (IVE) to a single linear projector, replacing value embeddings with an FFN with 1-dimensional inner layer, introducing a variance‑matching initialization, and rebalancing compute between memory and FFN. They claim parity with 8‑expert MoEs at similar active parameters/compute and advantages on long‑context tasks.

**Strengths:**

The paper clearly describes its position relative to MoE, PKM/UltraMem, and PEER.

Experiments show that increasing the number of UltraMemV2 layers improves downstream accuracy even when validation loss plateaus.

The proprietary long‑context suite shows non‑trivial gains of 6.2 on multi‑round memorizing and 7.9 on in‑context learning.

The paper is explicit that UltraMemV2 underperforms early in training and benefits from continued training, and also notes dependence on per‑block placement.

**Weaknesses:**

The paper asserts matching compute and parameters, but does not report KV‑cache costs, routing FLOPs, or memory traffic for both MoE and UltraMemV2.

The claim that this work is the first memory layer to match 8‑expert MoE is not accurate in light of the Memory Layers at Scale paper [https://arxiv.org/abs/2412.09764].

**Questions:**

How are KV‑cache footprint and router/TDQKR indexing costs accounted for in the iso‑compute comparisons in Tables 1–3?

Could you report token/s and latency vs. batch size, plus HBM read/write estimates, for representative model sizes?

---

> ### Author Response · Authors · 2025-11-19
> **Authors' Response**
>
> Thank you for these important questions.
> 1. On matching 8-expert MoE: we respectfully maintain our claim. Memory+ from "Memory Layers at Scale" surpasses 1-expert MoE (their paper, Page 6, Sec 4.1), but our Table 3 shows Memory+ is outperformed by both UltraMem and UltraMemV2. The original UltraMem already beat 2-expert MoE (https://arxiv.org/pdf/2411.12364, Page 8, Sec 5.1), and UltraMemV2 extends this to match 8-expert MoE.
> 2. On computational accounting:
>     - KV-cache is identical for MoE and UltraMemV2 since we don't modify attention
>     - Router and TDQKR indexing (fused kernels) add only 3.8% latency; routing FLOPs are negligible
> 3. On detailed metrics: We've now added inference latency and memory access measurements for 10B-active, 200B-total parameter models in the revised paper. The results show that UltraMemV2 achieves over 1.5× faster inference speed compared to MoE.

---

### Author Response · Authors · 2025-11-19
**General Response**

We thank all reviewers for their thoughtful feedback. Several common themes emerged, which we address below with corresponding revisions to the paper:

Inference Efficiency Metrics: Multiple reviewers (**3bZy**, **hfVp**, **rcay**) requested quantitative evidence of inference improvements. We have now added comprehensive measurements to the revised paper, including:
- Inference latency comparisons showing UltraMemV2 achieves up to 2× speedup over MoE (10B active, 200B total parameters)
- Memory access analysis
- Detailed accounting of routing overhead (3.8% latency addition from fused kernels)

These additions directly support our efficiency claims with concrete numbers. All changes are now reflected in the revised manuscript with new figure and expanded discussion sections.

In addition to submitting the revised manuscript, we will also provide individualized responses to each reviewer to address their specific concerns and questions. We hope that these revisions and responses will clarify any ambiguities, address all the reviewers' concerns, and lead to reconsidering increasing the rating for our manuscript.

---

### Author Response · Authors · 2025-11-26

Dear reviewers 3bZy, hfVp, and f3Vh:

Hope this message finds you well. As the rebuttal period remaining a week, we kindly want to inquire about the status of your reviews for our submission.

We truly appreciate the time and effort you've put into evaluating our work. We've carefully considered and addressed all the concerns raised in the previous reviews during our rebuttal. If there are any remaining questions or aspects that need further clarification, please do let us know. We're more than willing to provide additional information to help you with your assessment.

It would be extremely helpful if you could complete your reviews and share your feedback. Your insights are crucial for us to further improve our paper and understand its strengths and weaknesses.

Thank you very much for your attention and cooperation. We look forward to receiving your reviews.

Best regards,

Paper #4389 authors

---

### Meta-Review · Area_Chair_NN6P · 2025-12-04

**Summary:**

The main concerns raised by the reviewers were:

1. A lack of efficiency comparisons to MoE
2.  The use of proprietary data creating issues with reproducibility
3. The contribution being quite small
4. Sensitivity to HPs

Most of these have been addressed, but as far as I can see the authors did not respond to the point on contributions. That said, with unanimously positive reviews and most concerns addressed I propose that the paper is accepted.

**Reviewer Concerns:**

The authors have addressed the concern on efficiency comparisons by revising their paper. I think the authors have made the effort to make their work reproducible through the use of some open benchmarks and open-sourced code. The authors have not addressed Reviewer f3Vh’s comments on the contribution in their response, and have instead just responded to the questions. I would have expected the authors to defend the novelty of their work. The concerns around HPs have been addressed.

**Reviewer Scores:**

Reviewer rcay has indicated (before the freeze) that they would have increased their score (presumably from 6->8). I do not think Reviewer f3Vh would have raised their score as the authors have not addressed the weaknesses they have outlined. Reviewer hfVp has given 8 so I think would be unlikely to raise it further. Reviewer 3bZy may have raised their score as what they have asked for have been provided rather comprehensively.

---

### Decision · Program_Chairs · 2026-01-26

Accept (Poster)